# Evaluation of stress-controlled high-cycle fatigue characteristics in PLA-wood fused deposition modeling 3D-printed parts under bending loads

Morteza Kianifar[1], Mohammad Azadi[1]*, Fatemeh Heidari[2]

**1** Faculty of Mechanical Engineering, Semnan University, Semnan, Iran, **2** Department of Materials Engineering, Yasouj University, Yasouj, Iran

* m_azadi@semnan.ac.ir

**Data Availability Statement:** All relevant data are within the paper.

**Funding:** The author(s) received no specific funding for this work.

## Abstract

PLA (Poly-lactic acid)-wood provides more biodegradability through natural fibers, a significant advantage of pure PLA. Nevertheless, these bio-composites may have inferior mechanical properties compared to non-degradable polymer composites, considering the lower strength of natural particles compared to synthetic fibers. This research examines the fatigue behavior of additive-manufactured biopolymer PLA-wood and assesses its comparability with pure PLA. Therefore, solid fatigue test samples were printed using the FDM (fused deposition modeling) method. Afterward, fully reversed rotary bending fatigue experiments were performed at 4 different stress levels (7.5 to 15 MPa) to extract the S-N curve of PLA-wood. Moreover, the fatigue fracture surfaces of the PLA-wood were investigated and compared at the highest and lowest stress levels using an FE-SEM (Field Emission Scanning Electron Microscopy), indicating more ductile fracture marks at higher stress levels. The fatigue lifetime of the PLA-wood decreased by 87.48% at the highest stress level (15 MPa), rather than the result at the lowest stress level (7.5 MPa). Additionally, the results demonstrated that the fatigue characteristics of the printed pure PLA and PLA-wood were comparable, suggesting that the 3D-printed PLA-wood with the used printing parameters can be an alternative choice.

## 1) Introduction

Additive manufacturing (AM) technology allows the manufacturing of parts with complicated geometries from a CAD or computer-aided design model. FDM or fused deposition modeling, which is also known as the fused filament fabrication (FFF), is a prevalent AM method that operates by layer-by-layer extruding of a heated filament [1]. The AM is determined by various factors set manually by users in 3D printing machines. Therefore, the mechanical characteristics of components produced by the FDM method can be inexact [2, 3].

**Competing interests:** The authors have declared that no competing interests exist.

The mechanical behavior of 3D-manufactured polymers and their fatigue properties have been extensively examined recently. Studies revealed that various factors, such as material, the orientation of the print, thickness of the layers, the diameter and temperature of the nozzle, printing speed, and infill percentage, can affect the material characteristics of 3D-printed items [4–9]. In this case, Letcher and Waytashek [4] investigated the impact of the orientation of printing on the PLA fatigue lifetime and its tensile properties. They found that samples printed at 90˚ had the shortest fatigue lifetime. Moreover, Jerez-Mesa et al. [5] explored different printing parameters. They presented that the filling percentage and diameter of the nozzle showed the most meaningful influence on the fatigue behavior besides the layer thickness. Afrose et al. [6], through examining the influence of the print orientations on the fatigue lifetimes of the PLA, demonstrated that printing at a 45˚ angle had led to the most extended fatigue lifetime. Additionally, Ezeh and Susmel [7] considered the effect of the infill percentage and orientation of printing on the fatigue lifetime of the PLA.

In addition, Azadi et al. [8] had a comparison of the fatigue lifetime in the PLA and ABS and investigated the print direction effect. They showed PLA has better fatigue performance rather than ABS. They also noted that the horizontally printed sample has a higher fatigue lifetime than that in the vertical direction. In a review study, Safai et al. [9] investigated the fatigue characteristics of the 3D-printed specimens. It was specified that the optimal arrangement of parameters for fatigue endurance was challenging because of the interaction of material features and 3D printing parameters. Therefore, more experimental and computational fatigue investigations should be conducted to comprehend how printing parameters and materials influence fatigue properties.

Growing environmental awareness has led to increased research focus on green composites. These composites have gained more popularity than traditional petroleum-based counterparts because of the non-toxic and biodegradable nature of green composites. Green composites have found diverse applications from aerospace to household sectors and have been distinguished by their biodegradability through natural fibers and biodegradable polymers, which provide a significant advantage. Nevertheless, the green composites may have inferior mechanical characteristics, compared to non-degradable polymer composites, considering the lower strength and stiffness of natural fibers, compared to synthetic fibers [10, 11].

Ahmad et al. [12] studied enhancing the performance of 3D-printed bio-composite polymers, particularly focusing on natural fiber-based feedstock filaments by the FDM. That paper emphasized the need for a rheological analysis of these new composite materials derived from natural resources. It aimed to characterize their flow behavior and the optimal melting temperature for the extrusion process in the wire filament production. Moreover, Ahmad and Mohamad [13] optimized the accuracy of 3D-printed parts by applying the Taguchi method, considering variations induced by machine process parameters such as the quality, temperature, and speed. The comparison of coordinate measuring machine results with the CAD data, coupled with the Taguchi analysis, identified honeycomb print pattern, 0˚ of the y-axes orientation, 90˚ of the support angle, and 0.4 of the sidewalk offset as the optimum parameters. The print pattern exhibited the most significant impact on the part dimension accuracy, and the optimized parameters led to a 44.3% reduction in the shrinkage, improving the accuracy by 0.190 mm.

In the last few years, adding natural fibers as reinforcement for AM material has attracted interest, chiefly considering their sustainability properties. Previous research used additives and modifiers to the 3D printing material, chiefly nanoparticles and microparticles of wood, cellulose, bamboo, cork, flax, and other plant-based fillers [14–16]. These modifiers could be incorporated into a biodegradable thermoplastic material, such as PLA, utilized in FDM 3D printing [17–19]. Furthermore, improving mechanical properties can be considered an

additional advantage of kenaf-reinforced PLA, which has been shown to elevate its mechanical properties by 50% compared to pure material [20]. Daver et al. [17] indicated that the ductility of cork-reinforced PLA has grown. Additionally, their mechanical properties reduced while the percentage of cork increased and to have an equalized result, up to a maximum of 5% cork must be incorporated. Gkartzou et al. [21] investigated bio-based blends involving low-cost kraft lignin and PLA. Their findings indicated that the impact of reinforcement was contingent on the length-to-diameter ratio of particles. Specifically, they observed that pine lignin dust could create heterogeneous systems with PLA; however, the adhesion between the components might have a detrimental effect.

However, the complete understanding of the influence of reinforcements on the mechanical characteristics of 3D-manufactured materials remains an area that requires further exploration. Kariz et al. [22] observed enhancement in the tensile property of PLA-wood with an addition of 10% wood. At the same time, if that amount was higher, an adverse effect was observed, indicating the characterized specific materials for specific purposes were required.

PLA-wood is a composite material of polylactic acid and wood flour, a biodegradable and renewable alternative to petroleum-based plastics, and FFF can process it. Therefore, studying the fatigue performance of PLA-wood is critical for improving its performance and durability. The material properties and fatigue behavior of PLA-wood composites have been investigated narrowly. Tao et al. [23] studied the wood flour-filled PLA composite filaments for 3D printing through FDM. The study found that adding wood flour altered microstructure, enhanced initial deformation resistance, and had minimal impact on thermal properties. Tisserat et al. [24] tested the use of Paulownia wood flour (a bio-filler) for PLA. The study of resulting bio-composite mechanical properties reveals that wood particles enhance the stiffness of PLA while diminishing its flexibility. Guo et al. [25] investigated the utilization of PWF as a filler for FDM PLA composites, aiming to enhance their mechanical, impact strength, and interfacial properties. Ayrilmis et al. [26] confirmed that reinforcing polypropylene with walnut flour improved bending and tensile strength.

Travieso-Rodriguez et al. [27] performed a comparison of the rotary bending fatigue strength between PLA-wood composites and non-reinforced PLA specimens. It was shown that PLA-wood composites had greater fatigue strength than PLA specimens at low-stress levels but lower fatigue strength at high-stress levels. The findings suggest that PLA-wood composites exhibited lower endurance limits compared to PLA specimens because of wood particles, which increased the internal friction and energy dissipation and introduced defects and stress concentrations in the matrix. Muller et al. [28] examined the fatigue property of specimens made of natural PLA composites, such as cork, bamboo, and pinewood, using FDM technology. The fatigue tests and SEM analysis were used to measure the effect of cyclic stresses on the mechanical characteristics and fracture surface of the specimens. It was found that the natural fillers did not affect the fatigue life of PLA. In addition, samples exhibited increased visco-elastic behavior and permanent deformation due to cyclic loading. These studies provide valuable insights into the fatigue properties of PLA-wood composites manufactured by FFF. However, they also have some limitations and gaps that must be addressed in future research.

The significance of biodegradable PLA-wood composites is highlighted by their role in promoting environmental sustainability, reduced dependence on non-renewable resources, and their alignment with growing consumer and regulatory preferences for eco-friendly materials. These composites are suitable for applications with low mechanical stresses. At the same time, fatigue properties of PLA-wood need to be evaluated regarding pure PLA concerning potential reductions in fatigue life. Consequently, fatigue testing is essential to find the strength of PLA-wood composites under cyclic loading conditions.

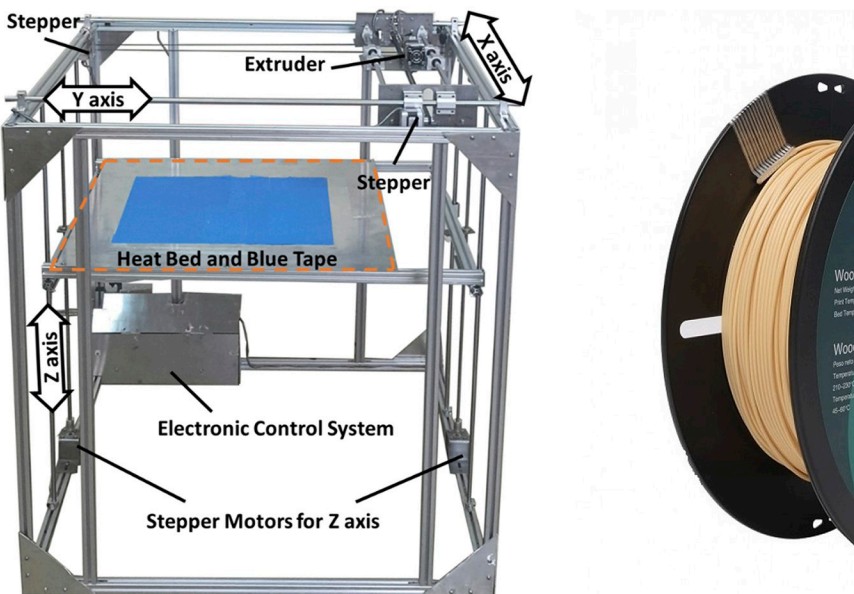
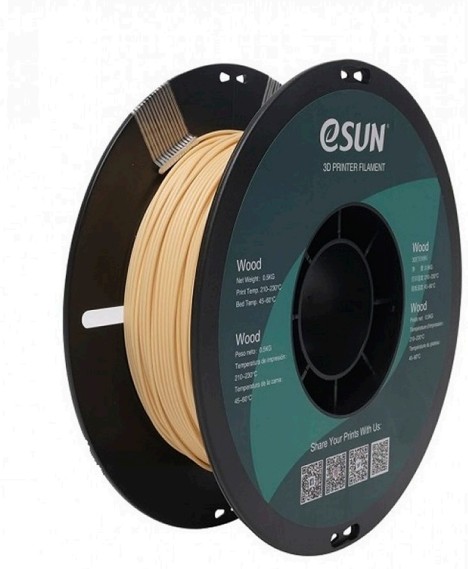

**Fig 1.** (a) The FDM 3D printer [31] and (b) the PLA-wood filament [29].

The literature review underscored the increasing utilization of PLA-woods and highlighted the significance of assessing their performance under cyclic loadings. Nonetheless, there has been limited exploration into the rotary bending fatigue behavior of the aforementioned material, while the obtained results were unfavorable. However, in this study the proposed printing parameters and the fatigue lifetime of the examined PLA-wood filament were found to be commendable, exhibiting the comparability with PLA. In this study, the S-N curve of the printed PLA-wood was extracted after conducting the fatigue tests. Then, results were compared with the pure PLA and indicated that the fatigue behavior of the studied biodegradable PLA-wood with used printing parameters was utterly comparable with pure PLA.

## 2) Research methods

In this research, the standard fatigue test samples were 3D-printed (FDM method) using the commercially available PLA-wood filament (1.75 mm) manufactured by Shenzhen eSUN Industrial Company. The percentage of the wood filler in the PLA matrix was 9% [29, 30]. These filaments were made by mixing the recycled wood particles and PLA to produce the effect of natural wood particles using starch raw materials obtained from plant-derived materials that can be replenished or regenerated naturally, which are eco-friendly and biodegradable.

Fig 1 depicts the FDM 3D printer device along with the utilized filament, and further details about the used filament properties are shown in Table 1. The FDM 3D printer device was

**Table 1. The properties of the used PLA-wood filament [29, 34].**

| Properties | Values |
|---|---|
| Density | 0.7 g/cm$^3$ |
| Melt flow index | 17 (190˚C/2.16 kg) |
| Melting point | 155–185˚C |
| Elongation at break | 12% |
| Ultimate tensile and yield strength | No data available |

**Table 2. 3D-printing parameters of PLA-wood samples fabricated with the FDM method.**

| Ref. | Material | Nozzle Temp. (˚C) | Bed Temp. (˚C) | Nozzle Diameter (mm) | Layer Height (mm) | Infill (%) | Infill Pattern | Print Speed (mm/s) | Print Direction |
|------|----------|-------------------|----------------|----------------------|-------------------|------------|----------------|--------------------|-----------------|
| [35] | PLA-wood | 240 | 90 | 0.4 | 0.2 | 100 | - | - | H |
| [36] | | 200 | 60 | 0.4, 0.6 | 0.15 | 100 | - | 50 | - |
| [37] | | 200 | 80 | 0.4 | 0.05, 0.1, 0.2, 0.3 | 100 | - | - | H |
| [38] | | 180, 200, 220 | - | 0.4 | 0.1, 0.3 | 100 | - | 30, 40, 50 | H |
| [27] | | 180 | 50 | 0.5, 0.6, 0.7 | 0.2, 0.3, 0.4 | 25, 50, 75 | HC., Rect. | 25, 30, 35 | - |
| [39] | | 210 | 60 | - | 0.3 | 80 | HC. | 3 | - |
| [40] | | - | - | 0.8 | 0.2 | - | - | 30 | - |

*Abbreviation: "Rect" means Rectangular, "HC" means Honeycomb, and "H" means Horizontal.

designed and built in the Advanced Material Behavior (AMB) research laboratory (Semnan University, Semnan, Iran). It was developed with three axes: the extruder moves on the X-Y plane with a belt driving system, and the build platform moves on the Z axis with a lead screw driving system. It can print parts up to $550 \times 600 \times 700$ mm$^3$ with an accuracy of 50 μm on the X-Y axes and 40 μm on the Z axis. The printer uses a NEMA 17 stepper motor, a driver with adjustable micro step resolutions, and the Marlin firmware [31].

It must be mentioned that the used variables for the 3D-printing process have been chosen considering the literature review on the printing parameters for PLA-wood, as given in Table 2, based on the literature [35–40]. Additionally, the dimension of the standard fatigue test samples was chosen according to the ISO-1143 [32]; nonetheless, it was designated for fatigue testing of metallic materials using a rotating bar bending setup. The geometry of the test sample and the printed sample is shown in Fig 2. To assess the fatigue characteristics of PLA-wood, high-cycle fatigue (HCF) experiments were carried out using the SFT-600, a two-point rotating bending fatigue device manufactured by Santam Company. The fatigue experiments were done under the loading frequency of 100 Hz, at room temperature [33].

Fatigue tests were performed at four amplitude stress levels (7.5, 10, 12.5, and 15 MPa), with each test repeated multiple times (at least three repetitions) at each stress level to ensure the repeatability of the testing process. Supplementary details about fatigue testing were provided in the former work of the authors [8]. Eq 1 was employed for deriving the fatigue properties by fitting the curve on a logarithmic scale from the acquired S-N curves [41]

$$\sigma_a = \sigma'_f (2N_f)^b \tag{1}$$

In this context, the stress amplitude was denoted as $\sigma_a$ in MPa, $N_f$ signifies the number of cycles until failure, while $\sigma'_f$ and $b$ represent the coefficient and exponent of fatigue strength, respectively.

To contrast the fracture surface of the printed PLA-wood specimen, two tested samples at the lowest and highest stress levels (7.5 and 12.5 MPa) were investigated using the Field Emission Scanning Electron Microscopy (including Sigma 300-HV, Zeiss FE-SEM device, Germany). In addition, Fourier Transform Infrared Spectroscopy (FT-IR) was employed to compare the pure PLA and PLA-wood filaments using the FT-IR 8400S device.

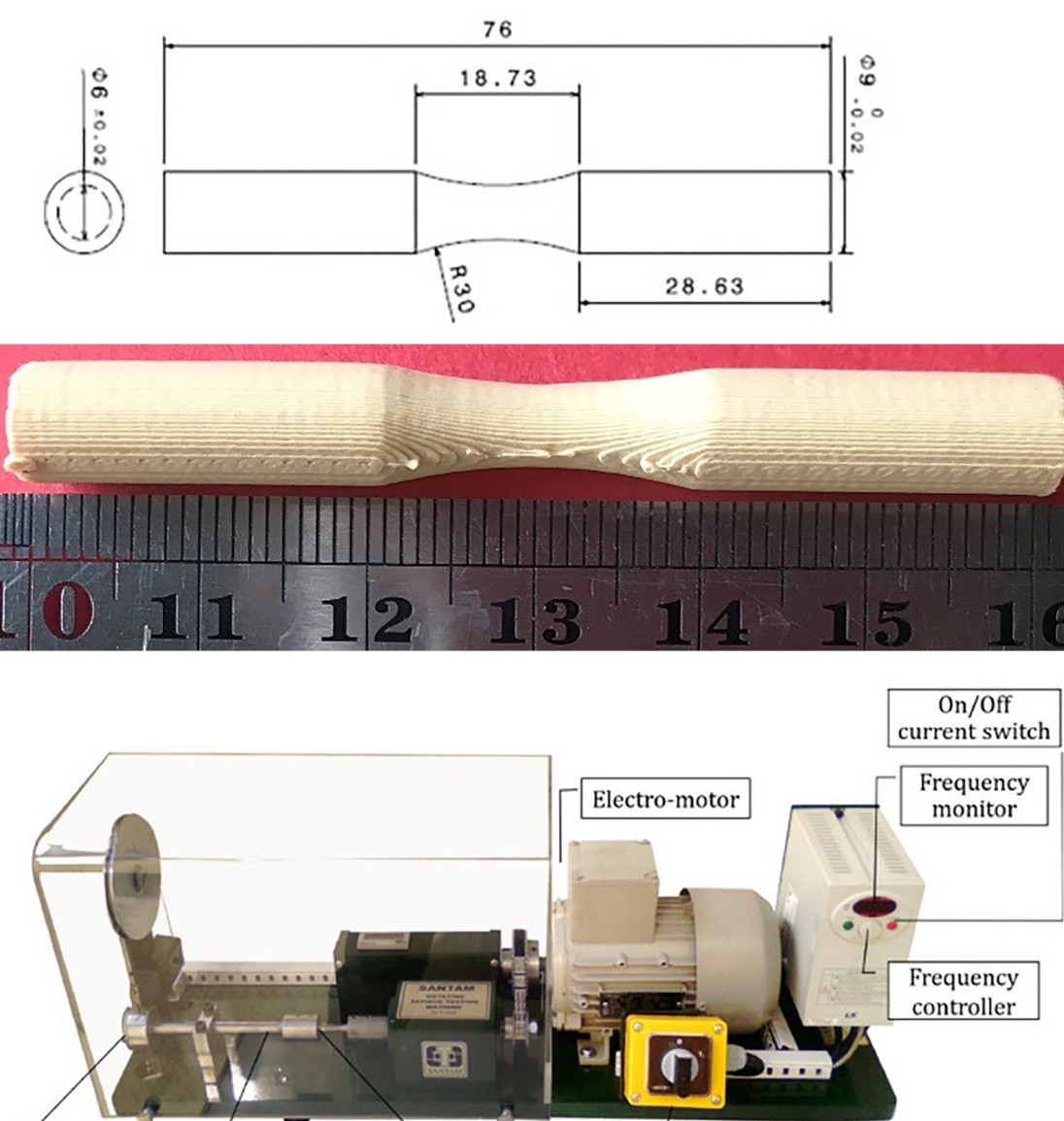

**Fig 2.** (a) The geometry map of fatigue specimen, (b) 3D-printed sample by the PLA-wood filament, and (c) the fatigue testing device [8].

## 3) Results and discussion

### 3.1) FT-IR spectroscopy investigation

The FT-IR examination was extensively utilized for identifying interactions and phase performance by recognizing the practical categories within the polymeric composites. The characteristic peaks in the neat PLA and PLA-wood composite are shown in Fig 3. The characteristic peak for neat PLA extracted at 3566, 3446, 2933, 2856, 1668, 1506, and 1087 $cm^{-1}$. As can be observed, the peaks at 1087 and 1668 $cm^{-2}$ were allocated to the stretching vibration of $C-O$ and $C = O$ groups, respectively. Moreover, the peak at 1506 $cm^{-1}$ was ascribed to the bending vibration of $-CH_3$ [42, 43]. The absorption peak at 2856 and 2933 $cm^{-1}$ was ascribed to an

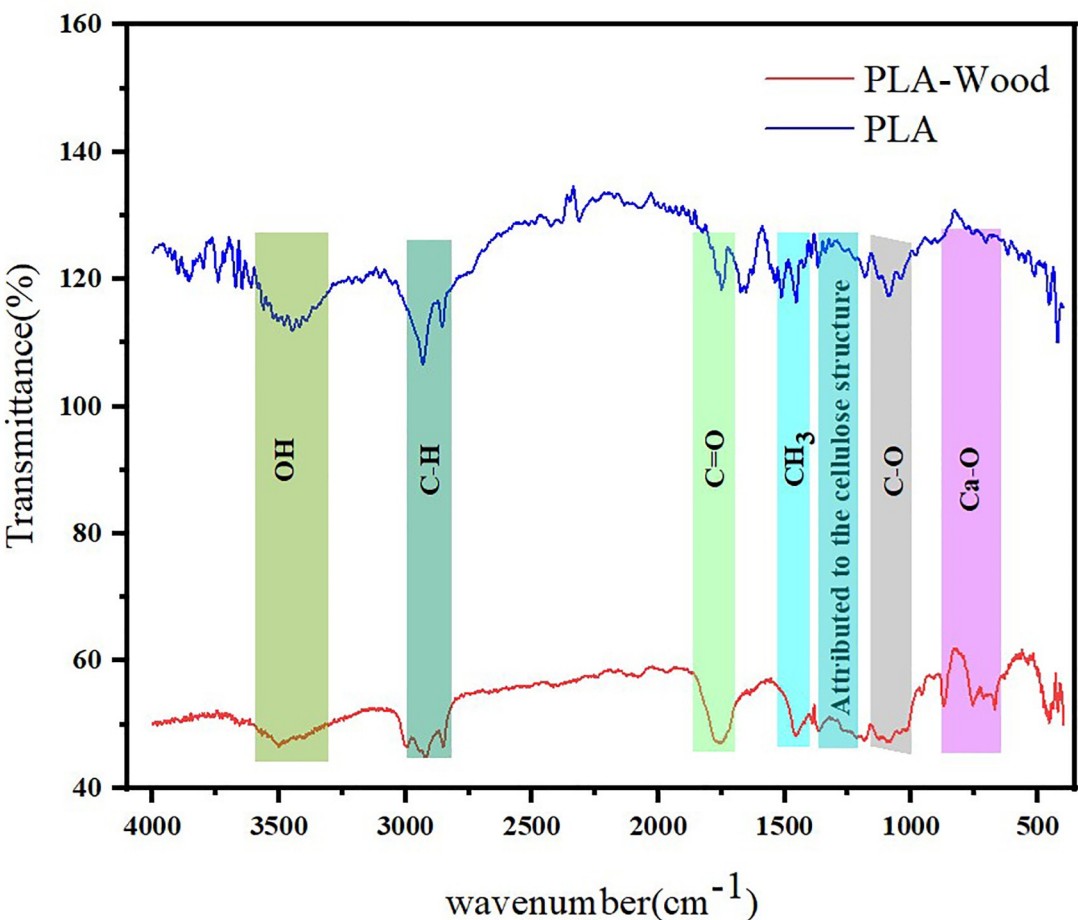

**Fig 3. The results of the FT-IR spectrum of pure PLA and PLA-wood composite.**

asymmetric stretching vibration by $-CH$, and the peak at 3446 and 3566 $cm^{-1}$ corresponded to a bending vibration mode by two terminal hydroxyl groups.

By adding the wood particles into the pure PLA, the absorption peak at 1087, 1668, and 3566 $cm^{-1}$ in the PLA, moved to new places with different wavenumbers (1097, 1751, and 3502 $cm^{-1}$). In addition, the bands at 1458, 1184, and 1097 $cm^{-1}$ were attributed to the cellulose structure, corresponding to $-CH_2$ symmetric bending, $C-O$ antisymmetric stretching and $C-O-C$ pyranose ring skeletal vibration, respectively. These variations in the spectra were supposed to exist based on the excellent interaction between the pure PLA and the wood particle filler through the construction of polar interactions among the functional groups of both constituents. Peaks at 756 and 871 $cm^{-1}$ belong to the asymmetric stretching vibrations of calcium-oxygen bonds of PLA-wood which is consistent with the chemical structure of PLA-wood filament according to the datasheet [29, 44].

## 3.2) Fatigue experimental data

The obtained S-N curve of the PLA-wood is shown in Fig 4. The fatigue lifetime of PLA-wood was compared to the fatigue behavior of pure PLA samples produced with various printing parameters used in the previous research conducted at the AMB research laboratory, emphasizing the excellent comparability of the PLA-wood 3D-printed samples.

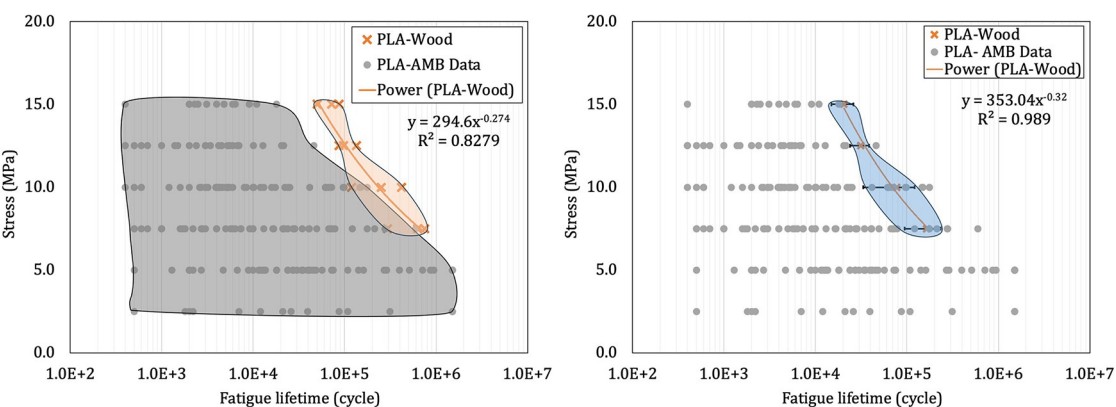

**Fig 4.** The S-N diagrams for PLA and PLA-wood samples (a) all data and (b) average data.

Moreover, the mechanical properties of the PLA filament used in AMB research laboratory (from the YouSu Company), and the PLA filament produced by the eSUN Company were compared in Table 3.

The printing parameters of the pure PLA and PLA-wood samples were contrasted in Table 4. In this case, the printing direction was horizontal and the infill pattern was rectangular. Fig 4(A) and 4(B) were generated using the complete set of fatigue data and the average values obtained from the fatigue tests at each stress level. It can be observed that at each stress level, the fatigue lifetime of PLA-wood samples is greater than that of all PLA samples that were made with different print parameters. Furthermore, the fatigue properties obtained based on each method are shown in Table 5. Indeed, the coefficient of determination or R-squared parameter would exhibit a higher value for the trend line of the averaged values, signifying the superior fitness of the proposed model regarding the all-data model.

**Table 3. The mechanical properties of PLA filament produced by the YouSu company and the eSUN company.**

| Mechanical property | YouSu Company [45] | eSUN Company [46] |
|---|---|---|
| Density (g/cm$^3$) | 1.24 | 1.24 |
| Melt flow index (g/10min) | 7 (210°C/2.16kg) | 5.2 (190°C/2.16kg) |
| Tensile Strength (MPa) | 60 | 60 |
| IZOD Impact Strength | 16 (J/m) | 4.3 (kJ/m$^2$) |

**Table 4. 3D printing parameters of pure PLA and PLA-wood samples.**

| Ref. | Material | Nozzle temperature (°C) | Bed temperature (°C) | Nozzle diameter (mm) | Layer height (mm) | Infill (%) | Print speed (mm/s) |
|---|---|---|---|---|---|---|---|
| AMB Laboratory | PLA | 180, 210, 240 | 30 | 0.2, 0.4, 0.6 | 0.2 | 60, 100 | 5, 10, 15, 20 |
| Present Study | PLA-wood | 200 | 30 | 0.4 | 0.2 | 100 | 15 |

**Table 5. The fatigue properties of the PLA-wood according to all experimental data and their averaged values.**

| All Data | | | Averaged values | | |
|---|---|---|---|---|---|
| $\sigma_f'$ [MPa] | $b$ [–] | $R^2$ [%] | $\sigma_f'$ [MPa] | $b$ [–] | $R^2$ [%] |
| 294.60 | -0.274 | 0.828 | 353.04 | -0.320 | 0.989 |

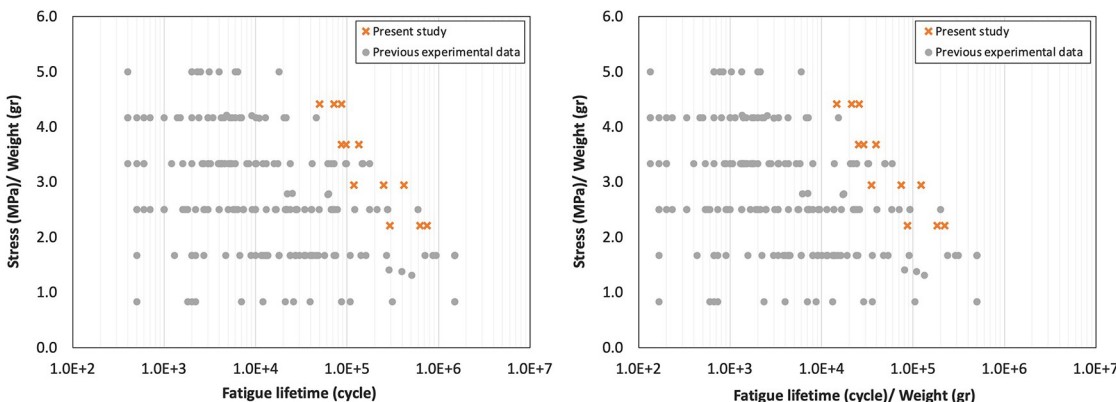

**Fig 5.** (a) The stress-to-weight ratio versus the fatigue lifetime and (b) the stress-to-weight ratio versus the fatigue lifetime-to-weight data.

Due to the substantial impact of infill on fatigue lifetime, strength, and weight in 3D-printed structures, it was advisable to mitigate the direct influence of the infill variable. A more effective approach was to analyze the ratio of material characteristics to the weight of the specimen, as also highlighted in the literature [47]. Therefore, to diminish the effect of weight and infill ratio, in Fig 5(A) and 5(B), the stress-to-weight ratio versus the fatigue lifetime and also, the stress-to-weight ratio versus the fatigue lifetime-to-weight data were drawn. The findings indicate that the fatigue lifetime of the printed biodegradable PLA-wood was comparable with the data for pure PLA and ranks among the best-obtained fatigue lifetimes.

The addition of wood to PLA can have various effects on its fatigue properties. Firstly, the inclusion of wood fibers often increases the overall stiffness of the PLA matrix. This issue leads to a composite material with improved rigidity compared to pure PLA. The increased stiffness can help the material resist the deformation and the fatigue failure under cyclic loadings [48, 49].

Additionally, the presence of the wood particles can influence the tensile strength of the composite. While the exact effect can depend on factors such as the size, shape, and distribution of the wood particles [50], in many cases, the tensile strength of the composite can be enhanced compared to pure PLA [23, 51]. For instance, Kain et al. [52] showed that the higher the fiber content in the filament can lead to better mechanical properties. Moreover, Kuciel et al. [53] studied the impact of wood and basalt fibers on the mechanical and hydrothermal properties of PLA composites and indicated the elastic modulus of composites improved by 45% with the addition of 15 wt.% wood Fibers.

However, it is important to note that the incorporation of the wood particles can also introduce directional dependencies in the mechanical characteristics of PLA. This means that the material properties may vary depending on the direction, in which it is tested. These directional dependencies can potentially affect the fatigue behavior of materials, as the distribution of stress and strain within the material can be influenced by the orientation of the wood particles. Overall, the addition of wood to PLA can result in a composite material with the improved stiffness, tensile strength, and potentially complex fatigue behavior due to the presence of directional dependencies.

### 3.3) Microscopic analysis

The creation of voids and micro-voids in polymers differs from that in metals. In metals, cavities in the neck region emerge post-necking due to geometric alterations that induce local

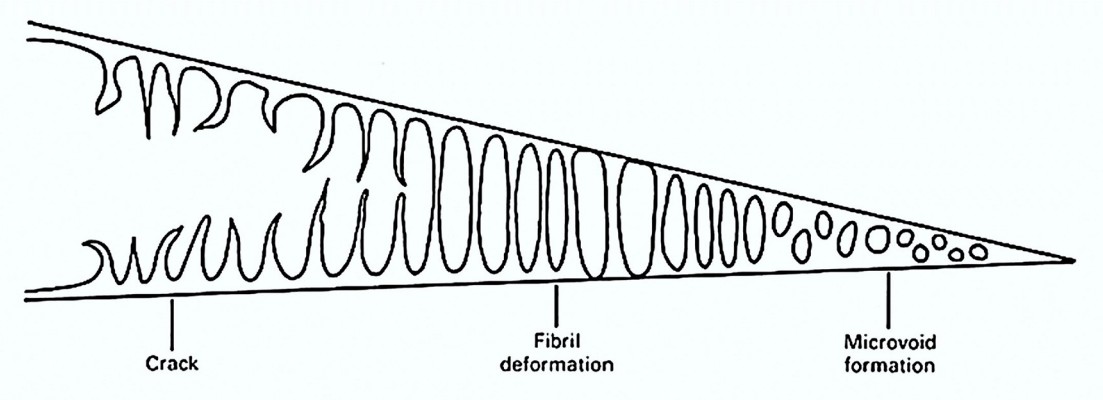

**Fig 6. Illustration of the formation of cracks stemming from craze marks [56].**

stresses during tensile loading. Micro-void accumulation leads to cavities and cracks, ultimately resulting in fractures [54, 55]. Cavitation processes in polymers can have a significant influence on plastic deformation. Unlike metals, these voids do not coalesce to create cracks; instead, they remain stabilized by fibrils that comprise directional polymers. Then, the micro-voids may continue growing into localized stress-whitening areas [56, 57].

Additionally, crazing is a prevalent feature in polymers, as depicted in Fig 6, and is typically characterized by three primary stages. Following micro-void formation, these micro-voids progress into fibrils before ultimately undergoing tearing [54]. In polymers, ductile fractures are characterized by considerable plastic deformation, whereas the brittle damage or fracture includes little or no macroscopically noticeable plastic deformations. Specifically, in polymers, the brittle damage or fracture arises at lower values of strains without substantial plastic deformations [56, 57].

Figs 7 to 9 compare the fracture surfaces of the highest stress level (15 MPa) and the lowest stress level (7.5 MPa) in various magnifications. The formation of the micro-void because of the plasticization in the fatigue fracture surface of PLA-wood was illustrated in Fig 8(D). The existing literature has documented the analogous morphology [58–60].

Figs 7 and 8 show that the fracture behavior can be classified according to stress levels. At the highest stress level (15 MPa), significant plastic deformation and fibrillation were evident across the fracture surface. Thus, the mentioned features, such as fibrillation, bridging, and severe deformation, are illustrated in Fig 7.

On the contrary, a different behavior can be observed in Fig 8 when subjected to the lowest stress level (7.5 MPa). Here, severe plastic deformation was mainly concentrated around the wood particles. This localized deformation directly results from stress concentration, where the material experiences intensified stress in specific regions [54]. Material discontinuities serve as stress concentrators, leading to the propagation of cracks from these points. This phenomenon causes local plastic deformations during cyclic loading, reducing mechanical properties and lessening the fatigue lifetimes [61–63]. This indicates a strong reaction of the material to changes in stress conditions, which leads to significant changes in its structural integrity. For a precise characterization of the various phases, the EDS point analysis of the fracture surface of PLA-wood is shown in Fig 10.

## 4) Conclusions

This study focused on investigating the high-cycle fatigue characteristics (under bending loads) of the PLA-wood biopolymer, comparing them with pure PLA. Cylindrical standard

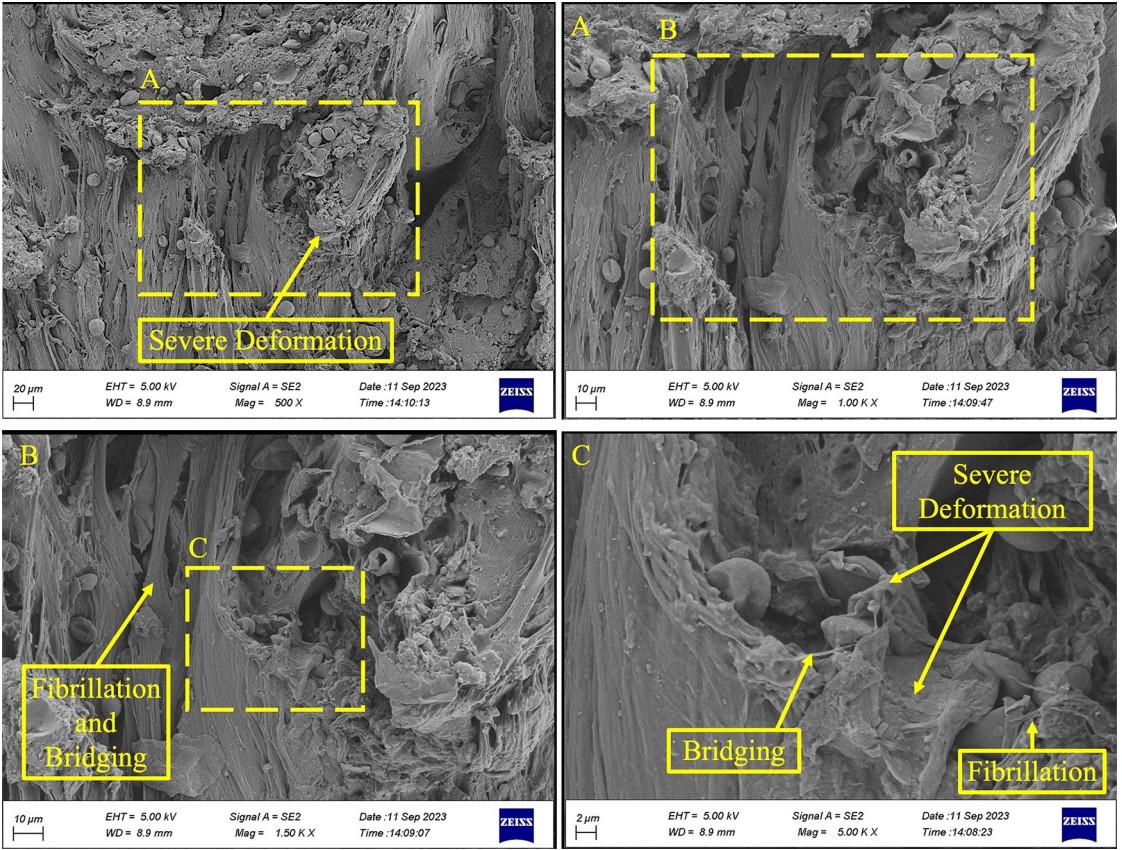

**Fig 7. The fracture surface images of the PLA-wood sample at the highest stress level (15 MPa).**

test specimens were fabricated using a fused deposition modeling 3D printer device. Subsequently, fully reversed rotary bending fatigue experiments were conducted at four different stress levels (ranging from 7.5 to 15 MPa) to establish the S-N curve for PLA-wood. The fracture behaviors of PLA-wood were then compared at the highest and lowest stress levels through FE-SEM analysis. The experimental results revealed:

- By adding the wood particles into the PLA, the PLA absorption peak shifted to new places with different wavenumbers. These variations are according to the proper interaction between the matrix (PLA) and the wood particles due to the construction of polar interactions.

- The obtained result indicates that the fatigue lifetime of used biodegradable PLA-wood was comparable with the data for pure PLA, which was made with different print parameters but also ranks among the best obtained fatigue lifetimes.

- The $\sigma'_f$ values for the studied samples were 294.60 and 353.04 MPa, based on all data and averaged values, respectively. Similarly, the $b$ values for the studied samples were -0.274 and -0.320, based on all data and averaged values, respectively.

- The PLA-wood exhibited a significant decrease of 87.48% in the fatigue lifetime when subjected to the highest stress level (15 MPa), compared to that of the lowest stress level (7.5 MPa).

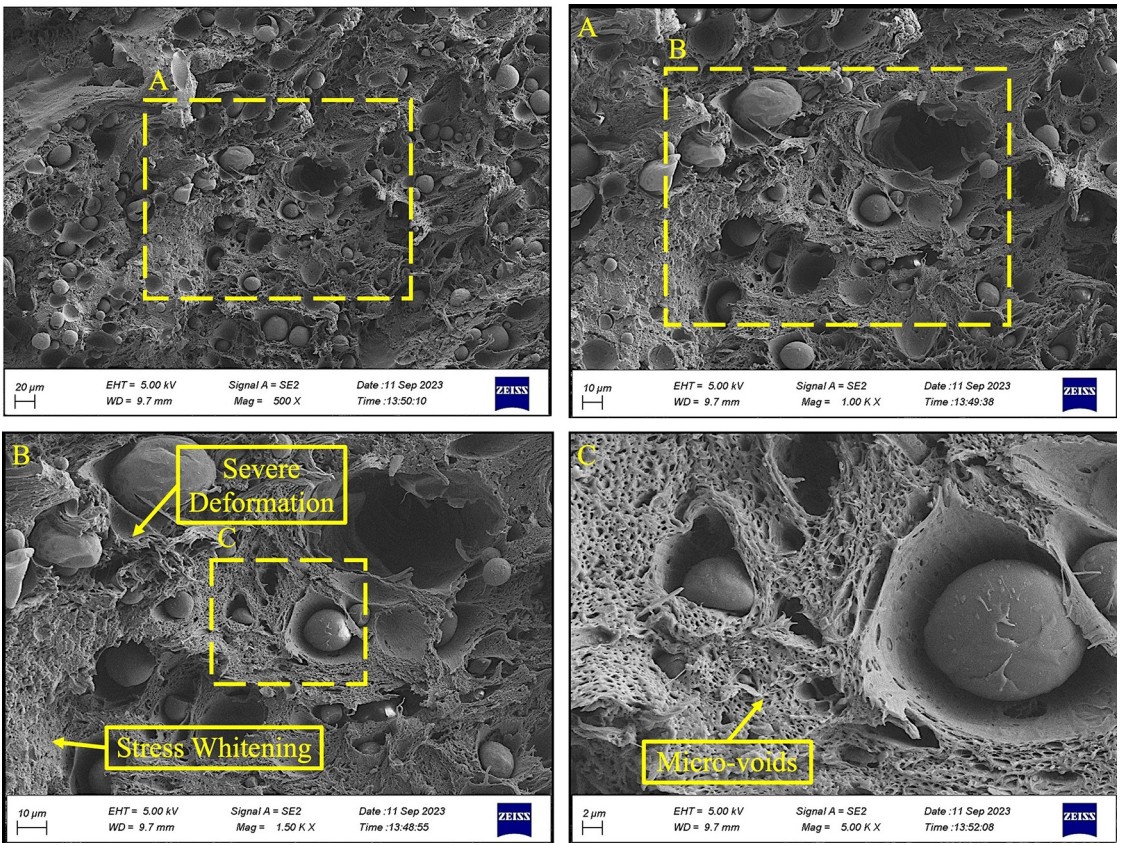

**Fig 8. The fracture surface images of the PLA-wood sample at the lowest stress level (7.5 MPa).**

- The existence of micro-voids, cavities, plastic deformations, and stress-whitening areas at the fracture surface indicated the ductile fracture behavior of the samples.

- Under the highest stress level (15 MPa), significant plastic deformation and fibrillation were evident across the fracture surface, while under the lowest stress level (7.5 MPa), severe plastic deformation was mainly concentrated around the wood particles. This localized

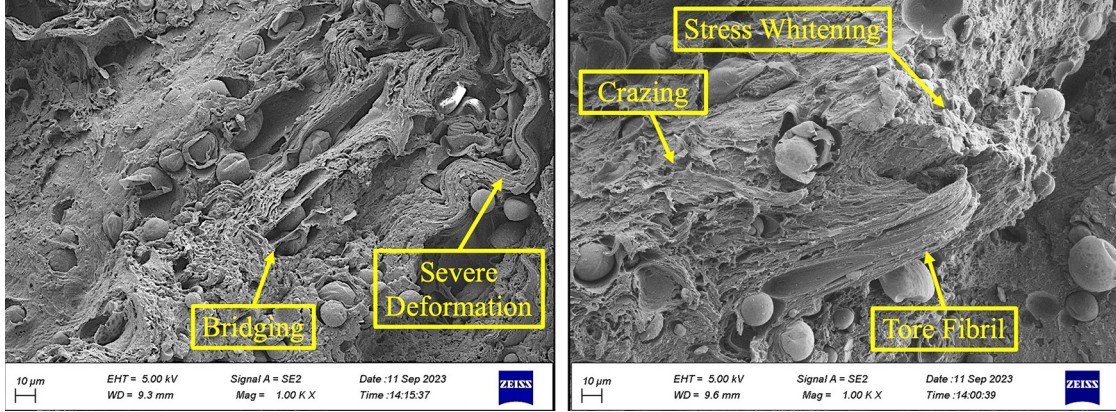

**Fig 9.** The fracture surface images of the PLA-wood sample at (a) the highest stress level (15 MPa), (b) the lowest stress level (7.5 MPa).

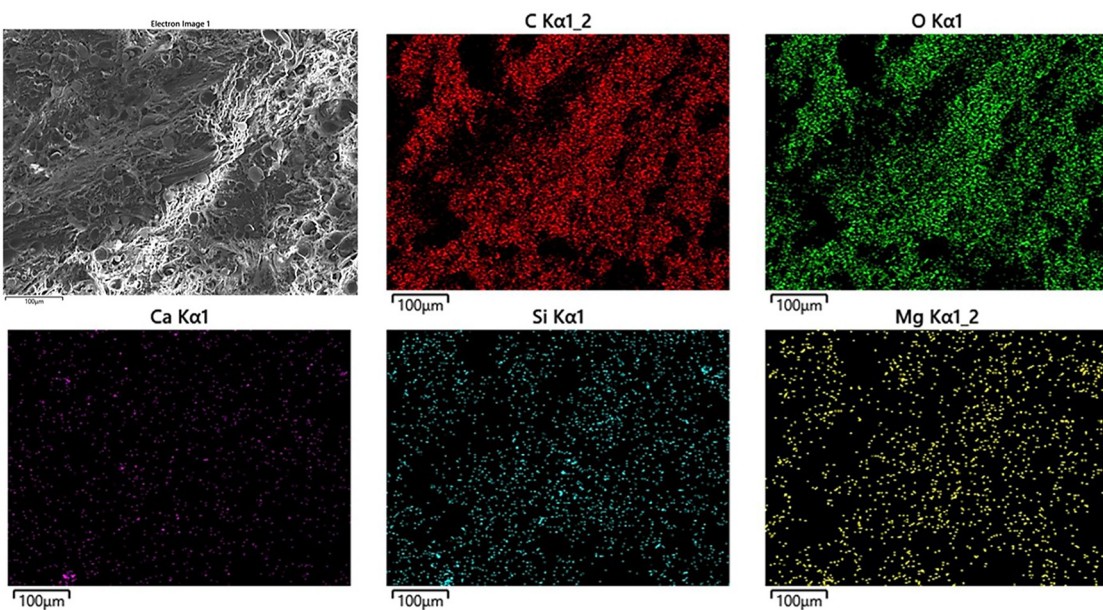

**Fig 10. The EDS point analysis of the fracture surface in PLA-wood.**

deformation directly results from stress concentration, where the material experiences intensified stress in specific regions.

For future investigations, considering the special attention in environmental fields and the increasing use of biopolymer composites in the field of biomedicine, improving the biological and mechanical characteristics of these materials is of particular importance. Therefore, conducting biomedical tests such as cell proliferation and corrosion and their effects on the fatigue lifetime of these materials can be the field of future studies. Furthermore, reducing the weight of these materials by using metamaterial structures while maintaining mechanical and biological properties is one of the other research topics in this field.

## Supporting information

**S1 Graphical abstract.**
(JPG)

## Author Contributions

**Conceptualization:** Mohammad Azadi, Fatemeh Heidari.

**Data curation:** Morteza Kianifar.

**Formal analysis:** Morteza Kianifar.

**Funding acquisition:** Mohammad Azadi.

**Investigation:** Morteza Kianifar, Mohammad Azadi, Fatemeh Heidari.

**Methodology:** Morteza Kianifar, Mohammad Azadi.

**Project administration:** Mohammad Azadi.

**Resources:** Morteza Kianifar, Mohammad Azadi, Fatemeh Heidari.

**Software:** Morteza Kianifar.

**Supervision:** Mohammad Azadi, Fatemeh Heidari.

**Validation:** Mohammad Azadi.

**Visualization:** Morteza Kianifar, Mohammad Azadi.

**Writing – original draft:** Morteza Kianifar.

**Writing – review & editing:** Mohammad Azadi, Fatemeh Heidari.

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
