## [Decision Letter · Decision Letter 0]

23 Jan 2024

PONE-D-24-00581Evaluation of stress-controlled high-cycle fatigue characteristics in PLA-wood fused deposition modeling 3D-printed components under bending loadsPLOS ONE

Dear Dr. Azadi,

Thank you for submitting your manuscript to PLOS ONE. After careful consideration, we feel that it has merit but does not fully meet PLOS ONE’s publication criteria as it currently stands. Therefore, we invite you to submit a revised version of the manuscript that addresses the points raised during the review process. Please submit your revised manuscript by Mar 08 2024 11:59PM. If you will need more time than this to complete your revisions, please reply to this message or contact the journal office at plosone@plos.org. Please include the following items when submitting your revised manuscript:A rebuttal letter that responds to each point raised by the academic editor and reviewer(s). You should upload this letter as a separate file labeled 'Response to Reviewers'.A marked-up copy of your manuscript that highlights changes made to the original version. You should upload this as a separate file labeled 'Revised Manuscript with Track Changes'.An unmarked version of your revised paper without tracked changes. You should upload this as a separate file labeled 'Manuscript'.

We look forward to receiving your revised manuscript.

Kind regards,

Khalil Abdelrazek Khalil, Ph.D.

Academic Editor

PLOS ONE

Journal Requirements:

Reviewers' comments:

Reviewer's Responses to Questions

**Comments to the Author**

1. Is the manuscript technically sound, and do the data support the conclusions?

Reviewer #1: Yes

Reviewer #2: Yes

2. Has the statistical analysis been performed appropriately and rigorously? 

Reviewer #1: Yes

Reviewer #2: Yes

3. Have the authors made all data underlying the findings in their manuscript fully available?

Reviewer #1: Yes

Reviewer #2: Yes

4. Is the manuscript presented in an intelligible fashion and written in standard English?

Reviewer #1: Yes

Reviewer #2: No

5. Review Comments to the Author

Reviewer #1: The authors study the evaluation of stress-controlled high-cycle fatigue characteristics in PLA-wood fused deposition modeling 3D-printed components under bending loads. The manuscript had an interesting topic and was well-written, however it could only be accepted with the following corrections:

1. For the title, it is suggested to change ‘3D-printed components’ to ‘3D-printed parts’. The word "components" might lead readers to get confused.

2. In abstract: a) ‘Therefore, Solid fatigue test’- ‘solid’ should be lower case. b) it is better to provide a scientific value or data of the result.

3. FFF (fused filament fabrication) can be mentioned, which is used on a par with FDM.

4. The content of the paper looks similar to previous review papers. Does this paper same with previous studies? If not, please highlight the differences. What is ‘wow’ factors in this paper?

5. Research gap was not found in the text (last paragraph from the introduction).

6. It is possible to cite the following papers in introduction section as references:

• Analysis on dimensional accuracy of 3D printed parts by Taguchi approach. Advances in Mechatronics, Manufacturing, and Mechanical Engineering. Lecture Notes in Mechanical Engineering, pp. 219-231 (2021), Springer Singapore.

• Rheological Properties of Natural Fiber Reinforced Thermoplastic Composite for Fused Deposition Modeling (FDM): A Short Review. Journal of Advanced Research in Fluid Mechanics and Thermal Sciences 98(2) (2022): 157-164.

7. Highlight the model of 3D printer and specification.

8. The authors used a commercial filament, PLA/wood. What is the composition of the wood?

9. Figure 4, why the number of samples between PLA and PLA/wood were unbalance or unequal?

10. The conclusion could be improved by adding the limitations and implications for researchers.

11. The authors should update the reference list (latest references from the year of 2019-2024)

12. There were a few typos in the text, which should be paid attention to (please proofread).

Reviewer #2: This article investigates the flexural fatigue behavior of FDM-3D printed PLA-wood composites. The following issues need to be addressed before publication:

1. The manuscript contains grammatical issues that need correction throughout the text.

2. Enhance the abstract by emphasizing key quantitative results and findings.

3. Given the absence of a 3D-printed pure PLA sample in the current study, exclude information related to pure PLA 3D printing material from Table 2.

4. Authors should explicitly state the equivalence of the mechanical properties of the PLA employed in the PLA-Wood filament from Shenzhen eSUN with the mechanical properties of PLA printed at the AMB laboratory. If the grades and properties match, a comparative analysis of their fatigue data can be conducted.

5. Revise the caption for Table 4 to clarify that it presents fatigue parameters resulting from regression analysis for all composite PLA-wood data, with no correlation to the fatigue parameters of PLA.

6. Referring to Figure 5, the results of the fatigue life to weight ratio for PLA-wood have been compared with all the results of fatigue life to weight ratio for PLA under various printing conditions. It is advisable to present the comparison of fatigue life to weight ratio between PLA-wood and PLA under consistent printing conditions in a tabular format. This will allow for a more accurate and quantitative comparison of these results under similar printing conditions.

7. Provide a more detailed analysis and interpretation of why 3D-printed PLA-wood samples exhibit superior fatigue life compared to PLA under various printing conditions.

8. To achieve a more accurate identification of different phases in the fractured cross-sectional images, consider conducting an EDS point analysis for precise characterization of the various phases.

6. PLOS authors have the option to publish the peer review history of their article (what does this mean?). If published, this will include your full peer review and any attached files.

Reviewer #1: No

Reviewer #2: No

---

## [Author Response · Author response to Decision Letter 0]

28 Feb 2024

Please check the file for the answers to comments.

---

## [Decision Letter · Decision Letter 1]

1 Mar 2024

Evaluation of stress-controlled high-cycle fatigue characteristics in PLA-wood fused deposition modeling 3D-printed parts under bending loads

PONE-D-24-00581R1

Dear Dr. Azadi,

We’re pleased to inform you that your manuscript has been judged scientifically suitable for publication and will be formally accepted for publication once it meets all outstanding technical requirements.

Kind regards,

Khalil Abdelrazek Khalil, Ph.D.

Academic Editor

PLOS ONE

Additional Editor Comments (optional):

Reviewers' comments:

Reviewer's Responses to Questions

**Comments to the Author**

1. If the authors have adequately addressed your comments raised in a previous round of review and you feel that this manuscript is now acceptable for publication, you may indicate that here to bypass the “Comments to the Author” section, enter your conflict of interest statement in the “Confidential to Editor” section, and submit your "Accept" recommendation.

Reviewer #1: All comments have been addressed

Reviewer #2: All comments have been addressed

2. Is the manuscript technically sound, and do the data support the conclusions?

Reviewer #1: Yes

Reviewer #2: Yes

3. Has the statistical analysis been performed appropriately and rigorously? 

Reviewer #1: Yes

Reviewer #2: Yes

4. Have the authors made all data underlying the findings in their manuscript fully available?

Reviewer #1: Yes

Reviewer #2: Yes

5. Is the manuscript presented in an intelligible fashion and written in standard English?

Reviewer #1: Yes

Reviewer #2: Yes

6. Review Comments to the Author

Reviewer #1: The authors used FDM to investigate the fatigue properties of PLA/wood composites. It sounds like a noteworthy result that might give impact to other researchers. The authors were able to reply all comments.

Reviewer #2: All review comments have been properly addressed in the manuscript, and the paper, in its current form, is recommended for publication in PLOSE ONE.

7. PLOS authors have the option to publish the peer review history of their article (what does this mean?). If published, this will include your full peer review and any attached files.

Reviewer #1: No

Reviewer #2: No

---

## [Editor Report · Acceptance letter]

4 Apr 2024

PONE-D-24-00581R1 

PLOS ONE

Dear Dr. Azadi, 

I'm pleased to inform you that your manuscript has been deemed suitable for publication in PLOS ONE. Congratulations! Your manuscript is now being handed over to our production team.

Kind regards, 

on behalf of

Dr. Khalil Abdelrazek Khalil 

Academic Editor

PLOS ONE